# A mobile intervention to reduce anxiety among university students, faculty, and staff: Mixed methods study on users' experiences

**Sarah Livermon**[1]*, **Audrey Michel**[2⊙], **Yiyang Zhang**[2⊙], **Kaitlyn Petz**[2], **Emma Toner**[2], **Mark Rucker**[1], **Mehdi Boukhechba**[1], **Laura E. Barnes**[1], **Bethany A. Teachman**[2]*

**1** Department of Systems and Information Engineering, University of Virginia, Charlottesville, United States of America, **2** Department of Psychology, University of Virginia, Charlottesville, Virginia, United States of America

⊙ These authors contributed equally to this work.
* sat2ew@virginia.edu (SL); bteachman@virginia.edu (BAT)

**Data Availability Statement:** Supplementary data has been deposited in Open Science Framework and can be found at https://osf.io/w8jzg/. All other

## Abstract

Anxiety is highly prevalent among college communities, with significant numbers of students, faculty, and staff experiencing severe anxiety symptoms. Digital mental health interventions (DMHIs), including Cognitive Bias Modification for Interpretation (CBM-I), offer promising solutions to enhance access to mental health care, yet there is a critical need to evaluate user experience and acceptability of DMHIs. CBM-I training targets cognitive biases in threat perception, aiming to increase cognitive flexibility by reducing rigid negative thought patterns and encouraging more benign interpretations of ambiguous situations. This study used questionnaire and interview data to gather feedback from users of a mobile application called "Hoos Think Calmly" (HTC), which offers brief CBM-I training doses in response to stressors commonly experienced by students, faculty, and staff at a large public university. Mixed methods were used for triangulation to enhance the validity of the findings. Qualitative data was collected through semi-structured interviews from a subset of participants (n = 22) and analyzed thematically using an inductive framework, revealing five main themes: Effectiveness of the Training Program; Feedback on Training Sessions; Barriers to Using the App; Use Patterns; and Suggestions for Improvement. Additionally, biweekly user experience questionnaires sent to all participants in the active treatment condition (n = 134) during the parent trial showed the most commonly endorsed response (by 43.30% of participants) was that the program was somewhat helpful in reducing or managing their anxiety or stress. There was overall agreement between the quantitative and qualitative findings, indicating that graduate students found it the most effective and relatable, with results being moderately positive but somewhat more mixed for undergraduate students and staff, and least positive for faculty. Findings point to clear avenues to enhance the relatability and acceptability of DMHIs across diverse demographics through increased customization and personalization, which may help guide development of future DMHIs.

data are in the manuscript and Supporting information files.

**Funding:** The following grants funded this work: the National Institute of Mental Health Smart and Connected Health 1R01MH132138-01 (BAT, LEB), the University of Virginia President and Provost's Fund for Institutionally Related Research (BAT, LEB, MB), and the University of Virginia Strategic Investment Fund Grand Challenge: Thriving Youth in a Digital Environment (BAT). The funders had no role in study design, data collection and analysis, decision to publish, or preparation of the manuscript.

**Competing interests:** The authors have declared that no competing interests exist.

## Author summary

Effective digital mental health interventions can help address the high levels of reported anxiety in college communities. These interventions can be used to reduce the barriers to accessing mental health services and promote flexible thinking to reduce the harmful impacts of anxiety. Despite evidence supporting the effectiveness of various DMHIs, their impact is often hindered by high attrition rates alongside low engagement. In this study, we gathered mixed-methods feedback data on a new DMHI called Hoos Think Calmly to assess individual user experiences that may impact engagement. Our goal is to better understand participants' evaluation of HTC and to gather feedback on how to enhance acceptability of the program in future iterations. Data was collected through feedback interviews and user experience questionnaires. We found that participants perceived HTC to be somewhat effective at reducing or managing their anxiety, but also noted barriers to ongoing use. Our study provides helpful insights into app use patterns, barriers to engagement, and general suggestions for improving DMHIs.

## Introduction

### Background

Anxiety disorders are highly prevalent among college communities. A 2023 study indicated that 41.8% of college students and 38.6% of faculty and staff had experienced symptoms consistent with severe or extremely severe anxiety over the past week [1]. Despite these alarmingly high rates, many individuals in need of mental health services remain untreated [2]. Some of the most common barriers to receiving mental health services among college students are access, time, and stigma. Accessibility barriers specifically include lengthy weight times, the cost of services, and perceived lack of options [3,4]. Similarly, university faculty reported stigma and lack of support as their main barriers to receiving mental health care [5].

In light of these barriers, there has been increased interest in digital mental health interventions (DMHIs). DMHIs provide mental health support via technologies like web-based or mobile platforms and have the potential to expand treatment delivery options and enhance access to mental health care, particularly in rural settings [6,7]. Additionally, studies show DMHIs may be specifically effective at reducing depression, anxiety, and stress outcomes in academic settings [8,9]. A recent meta-analysis [10] assessed the efficacy of digital interventions for various anxiety disorders, examining their impact against waitlist and standard care controls for specific disorders, and found positive effects on symptom reduction for DMHI's for generalized anxiety, mixed anxiety, and social anxiety disorders. However, it was noted that further research is necessary to investigate which intervention formats have the greatest impact on treatment outcomes, particularly regarding the type and frequency of support provided.

The use of Cognitive Bias Modification for Interpretation (CBM-I) is a mechanistic intervention that may be an effective component of treatment for anxiety [11–15], given extensive research showing that selective threat interpretations are common among individuals high in anxiety [16]. However, some studies report mixed results regarding the effectiveness of CBM-I [17,18], highlighting the need for further investigation into the efficacy of these programs and how to improve their acceptability. CBM-I training is designed to modify threat-focused cognitive biases related to interpretation processes (i.e., assigning threatening meanings to ambiguous situations). "Hoos Think Calmly" (HTC) is a mobile application

based on a specific CBM-I paradigm known as the ambiguous scenarios paradigm, which presents users with a series of ambiguous or mildly threatening scenarios [19]. In the typical ambiguous scenarios format, the final word of each scenario appears as a word fragment that participants must complete. These fragments are typically designed to assign benign or neutral interpretations of the scenarios in most cases, helping to shift participants away from selectively negative or catastrophic thinking patterns. Through repeated practice and reinforcement, the goal is to help individuals develop more flexible interpretation patterns that are less rigidly negative, ultimately reducing anxiety [20]. HTC offers these CBM-I training scenarios in brief 5–10 minute training sessions that are designed to increase flexible thinking in response to stressors commonly experienced by students, faculty, and staff at a large public university. To meet the specific needs of the university community, we engaged in outreach with student groups and organizations to beta test the app, review stressor domains, and develop scenarios tailored to their needs. We also consulted with leadership from local university programs for recommendations on in-app resources to incorporate.

Although there is evidence supporting the effectiveness of various DMHIs, such as online CBM-I training (including at our team's MindTrails website; [21–23]), their impact is hindered by high attrition rates alongside low engagement [24–26]. Better ways to overcome these common limitations of DMHIs need to be better understood. A 2021 review found that participants' engagement with DMHIs increased when users found the content appealing, perceived the program to be a good fit, and experienced perceived changes in their mental health [27]. Thus, we need to understand what makes content appealing and a good fit to users. Research on DMHIs tends to rely on quantitative metrics to evaluate outcomes, which are very valuable but limited in their ability to contextualize data and assess individual user experiences that may impact engagement. This points to the value of also using qualitative approaches to help design, evaluate and improve DMHIs [28–30]. For instance, qualitative methods can be used during the post-trial phase of pilot studies to gain insight into users' acceptance of and feedback on DMHIs [31]. Other studies have evaluated the acceptability of DMHIs through qualitative methods and have shown mixed feedback regarding attitudes toward and engagement with these tools. One study identified negative attitudes toward a Cognitive Behavioral Therapy (CBT) DMHI as users specifically found it to be inflexible, unrelatable, and hard to engage with [32]. Another study revealed that participants had a positive attitude toward and experienced benefits from a DMHI, but the need to further personalize and creatively engage users to promote long-term adherence was emphasized [33]. When it comes to enhancing user motivation, engagement, and enjoyment, recent research indicates that games and gamification, when combined with CBT, have produced encouraging results [34,35]. As the mobile health (mHealth) landscape continues to evolve to meet user needs, ongoing evaluation of DMHIs is essential to keep pace with the advancements in the field. This feedback can be especially valuable to guide improvements for future versions of a DMHI. Additionally, a 2022 paper emphasized the importance of distinguishing between the feasibility/acceptability of a DMHI and its effectiveness. It's argued that a program's efficacy cannot be accurately assessed if the intervention is not implemented properly and to its full potential [36]. Acceptability and feasibility studies help determine whether an intervention can meet users' needs and can be implemented as intended. This is essential for evaluating user interest and satisfaction, willingness to engage with DMHIs, factors influencing participation, and reasons for continuing or dropping out—all of which are important for refining the intervention, especially in its early stages. The findings from this study will offer concrete strategies for improving DMHIs based on user feedback, which can be used to enhance the acceptability of future interventions.

## Objective

The objective of this study is to collect feedback from participants in the first trial of a mobile application called "Hoos Think Calmly" (HTC) to evaluate acceptability and gain a deeper understanding of their experiences and suggestions for improvement. HTC is designed to increase flexible thinking in response to stressors commonly experienced by students, faculty, and staff at a large public university through brief ~5 minute interventions (microdoses) of CBM-I training delivered by mobile phone to university community members. Study design and analysis plans were preregistered on Open Science Framework [37].

## Methods

### Ethics approval and consent to participate

This study was approved by the Institutional Review Board for Social and Behavioral Sciences (IRB-SBS) at the University of Virginia. All participants provided written informed consent via an electronic consent form that outlined the study's purpose, risks, and benefits prior to study enrollment. To ensure confidentiality, each participant was assigned a unique ID after enrolling, and all data—including questionnaire responses and audio/video recordings—were securely stored and linked only to this ID, with access restricted to the research team.

### Strategies to understand and mitigate bias: Reflexivity and credibility

Our team acknowledges that we bring biases in favor of HTC, having grants that fund the program and prior tests of the parent program MindTrails that demonstrated efficacy in reducing anxiety [21–23]. It is likely that these factors may predispose us towards interpreting the current findings more positively.

To bolster methodological integrity, we employed several credibility strategies. Two of the initial coders on the research team were individuals not involved in the data collection process to help mitigate biases and allow for fresh perspectives [38]. Likewise, we established an inter-coder reliability score exceeding 0.80 on a subset of interviews before proceeding with general coding to increase coding consistency. Throughout the analysis, the lead author attended a recurring journal club focused on best practices in qualitative research. During this time, the lead author also spent time examining personal judgements and beliefs to better understand how biases might influence the research process and outcomes. We encouraged peer examination, and colleagues were asked to give feedback on our research methodology and procedures. An audit trail was kept of all edits made to the codebook throughout the analysis process to help establish confirmability and trustworthiness of data [39]. This study was reported using the Standards for Reporting Qualitative Research (SRQR) guidelines [40] (S1 Text).

### Intervention

In the parent randomized control trial (RCT), n = 134 participants assigned to the active treatment condition were notified to complete two microdose sessions per day over a 6-week period. Prior to beginning a microdose session, participants completed a pre-intervention survey in which they reported their current mood (and completed a very brief imaginal exercise depending on their anxiety level) and selected a specific stressor domain for targeted training. Domain options included academics/work/career development, social situations, romantic relationships, finances, physical health, mental health, family and home life, and discrimination. The content of the training scenarios was modified to match common situations encountered for the different groups (e.g., to capture fears of negative evaluation, the undergraduate group would read about giving a class presentation while the staff would read about presenting

at a staff meeting). Post-intervention surveys were also administered to assess changes in mood and provide additional resources and tips for emotion regulation and application of the CBM-I learning to daily life. At the end of the day, participants were instructed to complete a nightly survey inquiring about their mood and ability to think flexibly throughout the day. During this survey, participants had the option to pre-select notification times for the next day's microdose sessions.

A microdose session consisted of ~ 5 minutes of CBM-I training that aimed to reduce rigidly negative thinking patterns associated with anxiety. Training scenarios could consist of 10 CBM-I short scenarios (this occurred in the majority of sessions), or 1 long scenario, or 1 write-your-own scenario (except for the Discrimination domain, which did not include CBM-I) [41]. The variation was included to reduce repetitiveness and to introduce different tasks to increase flexible thinking and the personal relevance of training. A short scenario was typically three sentences in length and consisted of an emotionally ambiguous story that raised a potential threat but was then resolved when the user completed the final word in the last sentence. The emotional resolution was reinforced through a subsequent comprehension question (see Fig 1). A long scenario was the same as a short scenario except it also included a range of thoughts, feelings, or behaviors that could occur in a particular situation to help with seeing multiple different perspectives. Users were then prompted to write down their own thoughts that could be helpful in that situation. The write-your-own scenario invited participants to craft their own short CBM-I scenario based on a personally relevant anxiety-inducing situation. Users were then tasked with resolving the scenario in a positive or non-threatening way and coming up with reasons for why the chosen resolutions could be likely to occur. Fig 1 displays a screenshot of the app's homepage, domain selection page, and a short scenario example. More details on the intervention are available via Open Science Framework [41,42].

In the present study, some CBM-I scenarios were adapted from our existing pool of scenarios developed for previous MindTrails studies [21–23] and others were generated for the HTC

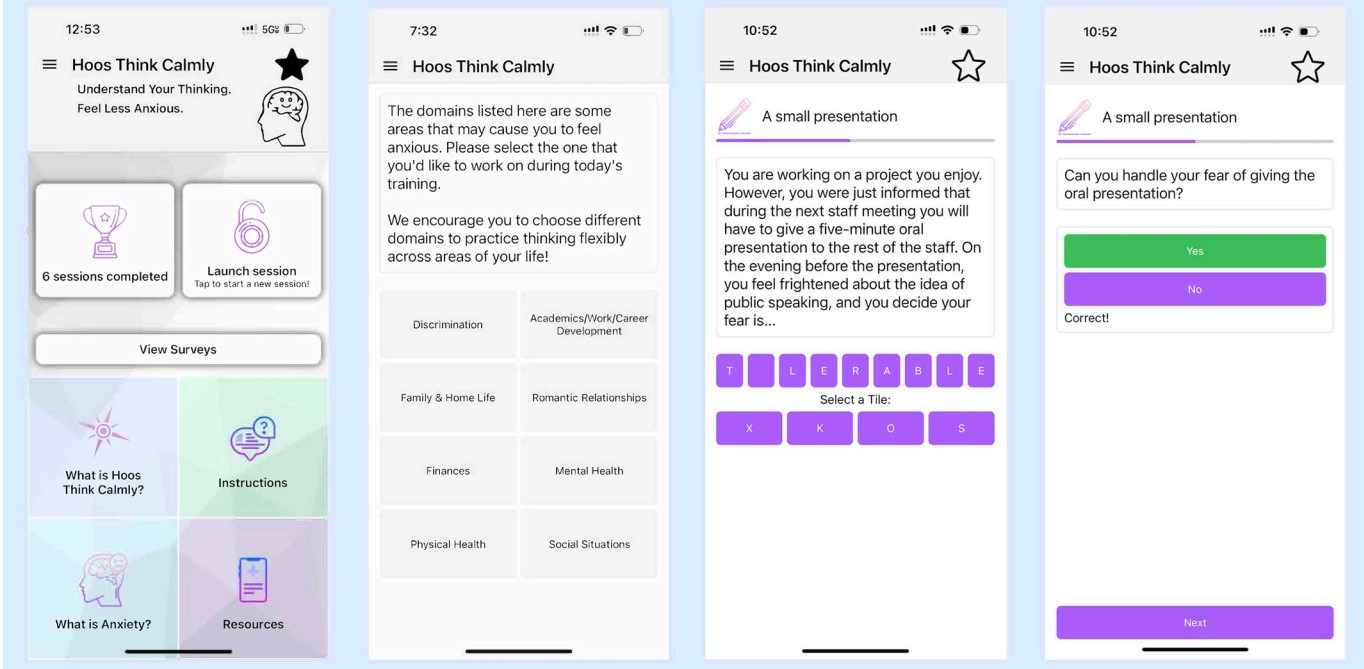

**Fig 1. Screenshots of the HTC app.**

application. Existing scenarios were adapted as needed to be relevant to university community members with different roles on campus and at different developmental stages. For example, a scenario pertaining to anxiety about handing in a paper to a teaching assistant might be kept the same (i.e., not adapted) for an undergraduate student, but modified to be focused on submitting a paper to an academic journal (graduate students) or giving an important report to a supervisor (staff members). Our goal was to preserve the core anxiety-provoking ambiguous stimulus for a scenario across groups but tailor the content to be more role-specific. This tailoring was guided in part by our qualitative interviews with university community members as well as in consultation with undergraduate students, graduate students, and faculty members on the research team who have lived experience as members of these groups. New scenarios were generated based on feedback from the qualitative interviews about topics identified as being particularly relevant for the university community. For example, we learned in feedback that many undergraduate and graduate students experience imposter syndrome and that this is a common source of stress and anxiety. We did not have any scenarios specifically pertaining to imposter syndrome in our existing pool, so we developed new scenarios focused on this topic. Scenarios were initially adapted and/or generated by a team of undergraduate research assistants who received training in how to write effective CBM-I scenarios. Subsequently, the HTC application was pilot tested, and scenarios were reviewed by other undergraduate students as well as graduate students and faculty in clinical psychology for small errors (e.g., typos) as well as for their effectiveness as CBM-I scenarios (i.e., whether the scenario vividly introduced a potentially threatening ambiguous situation and resolved that ambiguity in a positive or negative manner).

## Study procedures and data collection

This study used quantitative and qualitative approaches to understand user experience and acceptability of the HTC mHealth platform. Qualitative methods often provide greater depth in understanding, while quantitative methods are viewed as more objective and generalizable [43]. Mixed-methods analysis is often used to corroborate findings, elaborate on results, or generate new ideas [44]. We employed a mixed-methods approach to corroborate findings and enhance the credibility of the results through triangulation, while also elaborating on the results to achieve a more comprehensive analysis of the data [45]. While qualitative analysis was the primary focus of the study, quantitative methods were used to complement, confirm, and expand upon the qualitative findings. Data was collected via questionnaires and semi-structured interviews. Four biweekly user experience questionnaires were administered to all participants assigned to the treatment condition throughout their study participation. The first three questionnaires were administered every two weeks during the 6-week study period, and the final questionnaire was administered at the 8-week mark as a post-study follow-up. In addition to survey questionnaires, a subset of participants who took part in the 6-week pilot RCT to test the HTC mobile application were selected to participate in a user feedback interview. The selection process involved purposive sampling based on group assignment, study progress, and time passed since enrollment. Participants who were assigned to the treatment condition and had completed at least 3 microdose sessions were randomly selected to participate in a feedback interview. The semi-structured interview consisted of a mix of open-ended questions about user perceptions of and suggestions for the application, and more targeted questions that addressed topics that would be relevant for app design and content decisions. Questions were based in part on prior qualitative interviews conducted by our team and common user experience measurement tools [46]. In prior qualitative interviews, the semi-structured interview guide was iteratively updated to prioritize questions related to different design

decisions (e.g., earlier interviews focused on topics that should be covered in CBM-I scenarios, whereas later interviews focused more on developing a feasible implementation plan with respect to app notifications, session length, etc.).

The interview selection process was initiated by inviting participants with the closest study enrollment date to the time of recruitment and moving backwards until our target accrual number was met. We aimed to complete approximately 20 interviews based on norms in the literature for papers with similar goals [28,47] and also due to practical considerations. A total of 89 interview invites were sent to faculty members, staff members, graduate students, and undergraduate students, and 22 interviews were completed using a semi-structured interview guide (S2 Text). Participant characteristics are provided in Table 1.

**Table 1. Characteristics of interviewee participants and survey respondents.**

|  | Questionnaire sample (n = 95) | | Interview sample (n = 22) | |
| --- | --- | --- | --- | --- |
| Variable | n | % | n | % |
| **Role at university** | | | | |
| Undergraduate | 26 | 27.4 | 5 | 22.7 |
| Graduate | 31 | 32.6 | 10 | 45.5 |
| Faculty | 6 | 6.3 | 3 | 13.6 |
| Staff | 32 | 33.7 | 4 | 18.2 |
| **Gender/Sex** (the item did not distinguish between gender identity and sex at birth) | | | | |
| Woman | 76 | 80.0 | 18 | 81.8 |
| Man | 12 | 12.6 | 3 | 13.6 |
| Transgender Man | 0 | 0 | 0 | 0 |
| Transgender Woman | 0 | 0 | 0 | 0 |
| Other identity | 0 | 0 | 0 | 0 |
| No answer provided | 7 | 7.4 | 1 | 4.5 |
| **Race** | | | | |
| White/European origin | 63 | 66.3 | 13 | 59.1 |
| South Asian | 7 | 7.4 | 2 | 9.1 |
| East Asian | 7 | 7.4 | 2 | 9.1 |
| Black/African origin | 4 | 4.2 | 0 | 0.0 |
| More than one race | 1 | 1.1 | 2 | 9.1 |
| Other or Unknown | 6 | 6.3 | 2 | 9.1 |
| No answer provided | 7 | 7.4 | 1 | 4.5 |
| **Ethnicity** | | | | |
| Not Hispanic or Latino | 83 | 87.4 | 20 | 90.9 |
| Hispanic or Latino | 5 | 5.3 | 1 | 4.5 |
| No answer provided | 7 | 7.4 | 1 | 4.5 |
| **Measures at study baseline** | | | | |
| OASIS | Mean 6.39 (SD = 2.87) | | Mean 6.52 (SD = 2.42) | |
| PHQ-2 | Mean 1.63 (SD = 1.46) | | Mean 1.71 (SD = 2.42) | |
| PSS-4 | Mean 6.88 (SD = 3.10) | | Mean 7.48 (SD = 3.39) | |

OASIS = Overall Anxiety Severity and Impairment Scale; Scores range from 0–20. A higher OASIS score indicates more severe anxiety symptoms.

PHQ-2 = Patient Health Questionnaire-2; Scores range from 0–6. A higher PHQ-2 score indicates more severe depression symptoms.

PSS-4 = Perceived Stress Scale-4; Scores range from 0–16. A higher PSS-4 score indicates more stress.

SD = standard deviation

Patient Health Questionnaire-2 (PHQ-2) was used to assess depression symptoms. It consists of two 4-point Likert items, ranging from 0 (not at all) to 3 (nearly every day). A score of 3 or greater suggests the potential for major depressive disorder. Previous research supports the measure's construct and criterion validity, discriminative validity, and test-retest reliability [48–50]. The Overall Anxiety Severity and Impairment Scale (OASIS) was used to assess the frequency and intensity of anxiety symptoms. It consists of five 5-point Likert items, ranging from 0 (lowest value) to 4 (highest value), with higher scores indicating more frequent and severe anxiety. The OASIS has demonstrated strong psychometric properties, including good convergent validity and test-retest reliability [51–53]. Other MindTrails studies have also found acceptable internal consistency for this measure [21,23]. Perceived Stress Scale 4 (PSS-4) was used to assess stress through four 5-point Likert items, ranging from 0 (never) to 4 (very often). A higher score indicates more stress. Previous research has found acceptable internal consistency and reliability for this measure [54,55].

## Analysis

Following our pre-registration [37], interview data was analyzed using thematic analysis as outlined by Braun and Clarke [56]. All interviews were transcribed using Descript audio transcription service [57]. Each transcript was then checked against the original audio recording by a member of our research team. This was for the purpose of ensuring transcription accuracy and enhancing team familiarity with the data. During transcript review, team members noted initial themes found in the data.

After all interview transcripts had been reviewed, preliminary codes were discussed, and an initial codebook was created. We then used an iterative coding process, allowing for continual addition or deletion of codes to align with our data. To most effectively capture context and meaning, text was coded at the "level of meaning", allowing a code to encapsulate a line, sentence, or paragraph [58]. Interviews were coded using Delve qualitative data analysis software [59]. We followed an inductive approach of open coding to allow for unexpected, data-driven themes to emerge [47]. As codes were created, they were grouped with relevant codes to highlight similarities and begin developing overarching themes. These themes were identified and agreed upon through group consensus, although they continued to be refined as codes were added and removed throughout the analysis process. Four coders were trained by conducting inter-coder reliability (ICR) tests. Inter-coder reliability was measured using Krippendorff's alpha, a measure of reliability in content analysis [60]. We underwent six rounds of ICR testing and reached an ICR rating of 0.83. Each round of testing involved independent coding of the same transcript by each coder, followed by comparison and discussion of coding differences. Transcripts from different participant groups (faculty, staff, graduate, and undergraduate) were selected for each round of ICR testing to ensure representation of the entire data set [61]. Changes were made to the codebook throughout this process. Each codebook iteration, including the final version, can be found on Open Science Framework [62]. After an inter-coder reliability score greater than 0.80 had been established, official coding of the transcripts commenced. Due to prolonged ICR training, only two of the four trained coders were available for the official coding process. Given time constraints and the high ICR rating, deviations were made to the preregistration, with each transcript being coded by one of the two remaining coders instead of undergoing double coding. Based on norms in the literature, it is not uncommon for ICR to be calculated on a subset of data (10–25% of data units) to establish satisfactory reliability and then proceed to single coding [61,63]. In line with this approach, we double coded 27% (6/22) of our transcripts before establishing ICR and moving to single coding. However, we acknowledge that single coding may introduce limitations in analysis due to potential

inconsistencies in interpretation among independent coders, as well as increased bias from relying on a single analysis. To mitigate this, weekly meetings were held to address coding challenges and refine the codebook. The lead author refrained from coding to maintain an unbiased third-party perspective and acted as a moderator to help resolve disputes or clarify coding issues. In cases of disagreement, the lead author had the final say on coding decisions.

Thematic saturation was attained after analyzing 15 transcripts and was determined when no additional codes could be added to the codebook [64]. Once coding was complete, the themes were reviewed and refined one final time by visualizing their relationships with the codes and with each other through thematic mapping. The final codebook consisted of 47 main codes (not including sub-codes) that were grouped into 10 themes. An initial thematic map was created to visualize the codes and their relationships. Of those 46 codes, 13 appeared in 50% or more interviews. A revised thematic map was created using the 13 most saturated codes, and 5 main themes were assigned (S1 Fig). The full list of codes and definitions can be found in the codebook. Survey data was analyzed using descriptive statistics, including frequency counts and percentages.

## Results

The results are categorized based on the 5 themes found in Table 2, along with examples of deidentified illustrative quotes. Considering the primary research goal for this paper was learning users' impressions and suggestions from the qualitative feedback, and the survey data was included for triangulation purposes, we decided prior to the quantitative analyses to only examine those survey questions that aligned with the qualitative themes. Survey data are reported in Table 3.

### Theme 1: Effectiveness of training program

**Application of new coping strategies.**  Of the interview participants, 50% (11/22) expressed an ability to effectively apply new coping strategies (e.g. flexible thinking) that they learned from the program to their real-life situations. Notably, of these 11 participants, 8 of them were graduate students (73%), 2 were undergraduate students (18%), 1 was staff (9%), and none were faculty.

**Table 2.  Main Themes Identified and Frequency of Codes among Interview Participants.**

| Theme | Code | Total (n = 22) |
|---|---|---|
| Effectiveness of the Training Program | Application of new coping strategies | 11 (50%) |
| | Perceived changes in stress/anxiety levels | 14 (67%) |
| Feedback on Training Sessions | Relatability | 21 (95%) |
| | Repetitiveness | 12 (54%) |
| | Length | 18 (82%) |
| | Write-your-own | 14 (64%) |
| Barriers to Using the App | Lack of understanding of intervention format or rationale | 12 (55%) |
| | Forgetfulness | 12 (55%) |
| Use Patterns | Used routinely | 15 (68%) |
| | Used when reminded | 11 (50%) |
| | Chosen academics/work/career development domain | 18 (82%) |
| Suggestions for Improvement | Feature Suggestions | 19 (86%) |
| | Domain Suggestions | 16 (73%) |

**Table 3. Results from Questionnaires.**

| Survey question | 1 = Not at all | 2 = Slightly | 3 = Somewhat | 4 = Mostly | 5 = Very | Mean ± SD |
|---|---|---|---|---|---|---|
| 1. How helpful do you find Hoos Think Calmly for reducing your anxiety? | 35 (15.63%) | 57 (24.45%) | 97 (43.30%) | 30 (13.39%) | 5 (2.23%) | 2.61 ± 0.98 |
| 2. How much could you relate to the stories presented during training sessions? | 9 (4.31%) | 57 (27.28%) | 76 (36.36%) | 46 (22.01%) | 19 (9.09%) | 3.04 ± 1.02 |
| 3. How believable did you find the stories' endings in terms of those outcomes happening in your life? | 8 (3.83%) | 52 (24.88%) | 67 (32.06%) | 60 (28.71%) | 18 (8.61%) | 3.14 ± 1.02 |
| Survey question | 1 = Way too short | 2 = Somewhat too short | 3 = The right length | 4 = Somewhat too long | 5 = Way too long | Mean ± SD |
| 4. I find the [length of the] training sessions to be: | 0 (0.00%) | 1 (1.09%) | 46 (50.00%) | 41 (44.57%) | 2 (2.17%) | 3.49 ± 0.57 |

*Percentages do not add up to 100 when "Prefer not to answer" was endorsed

*There was this scenario that is about the advisor frowning in front of you and that instinctively made me think that I did something wrong or what I said is something stupid, but the answer was that people frown when they think about something, and that scenario really helped me to think differently when I present my work in front of other people or in front of advisors.*

*[Grad 13]*

**Perceived changes in stress or anxiety.** In addition, 64% (14/22) of interview participants discussed the program's impact on their stress or anxiety. The majority of those participants (8/14, 62%) expressed an overall positive change in symptoms, stating a reduction in their stress or anxiety levels.

*It [the program] definitely did help. . .I immediately go to the worst case scenario, so I thought it helped like in the heat of the moment for me to redirect my train of thought to somewhere that's more productive and less doomsday.*

*[Undergrad 35]*

Some of those participants (5/14, 38%) reported no change to their stress or anxiety while using the app.

*This just didn't do anything for me. It just added another thing in my day that I had to think about and take care of that didn't feel like it was giving me anything back.*

*[Undergrad 39]*

Additionally, two participants (2/14, 14%) expressed a negative change in symptoms, reporting an increase in their stress or anxiety levels from use of the app.

*Some [scenarios] seemed to cause me to worry more about things. Like I understand that it's like guiding you through stressful situations and it's not always gonna have a positive outcome for everything, but some of them were a little bit like, Ooh, that's kind of scary to think about.*

*[Undergrad 28]*

It is worth noting that one participant reported experiencing both positive and negative changes in their stress or anxiety levels. Therefore, there are a total of 15 responses indicating positive, negative, or no change, out of the 14 participants who commented on the program's impact.

Additionally, a survey questionnaire was sent to all participants at the 2nd, 4th, 6th, and 8th week marks of their study participation to assess the effectiveness of the HTC program in reducing their anxiety. A total of 225 responses were received (participants submitted responses on multiple occasions, hence the greater number of responses than participants), with the most common response (97/225, 43%) being that HTC was somewhat helpful at reducing their anxiety (see Table 3).

## Theme 2: Feedback on training sessions

**Relatability.**   Out of the 22 interview participants, 95% (21/22) discussed the relatability of the training scenarios. Among those 21 participants, approximately half (11/21, 52%) noted that the training scenarios resonated with their own experiences and feelings. Notably, of these 11 participants that found the program relatable, 8 of them were graduate students (73%), 2 were undergraduate students (18%), 1 was staff (9%), and none were faculty.

*I thought they were really [applicable], and sometimes I would even laugh because they would be so accurate that I'm like, yeah, I went through that like last week or so.*

*[Grad 49]*

*And I thought especially for academics, the situation or the stories that were there were quite relatable. Like I thought, yeah, this is exactly what I'm feeling right now or what I'm stressed about right now.*

*[Grad 35]*

Two thirds of participants (14/21, 67%) indicated that they sometimes did not find the training scenarios relatable.

*I felt like the demographic was younger than I am. It was a lot of you've got a new baby, you've got a young child, or you've got parents who disapprove of what you're doing. My parents are dead. This is not a thing I'm thinking about now. . . It just tended to not be applicable to me.*

*[Faculty 8]*

*There were times where I thought, oh, this doesn't relate to me at all, and why am I taking time to do this?*

*[Staff 39]*

Notably, 4 participants mentioned finding the training scenarios to be relatable at times while not at others, accounting for the 25 responses among the 21 participants. Additionally, it is worth mentioning that all 3 faculty interview participants found the training scenarios to be unrelatable.

Similar trends were observed in the user experience questionnaire. Participants were asked to evaluate the relatability of the stories as well as the believability of the stories' endings at the 2nd, 4th, and 6th week marks of their study participation, resulting in a total of 209 responses. The majority of participants provided responses within moderate ranges for both questions,

with 36% (76/209) finding the stories somewhat relatable and 32% (67/209) finding the endings somewhat believable (see Table 3). On average, faculty members tended to rate the scenarios as less relatable and the endings as less believable compared to the other participants.

**Repetitiveness.** Among the interview participants, 54% (12/22) found the training sessions to be repetitive. In certain aspects, repetition was intentionally incorporated into HTC as a way to give users repeated practice in thinking more flexibly and exposure to different scenarios. However, some found the repetitive writing style of the scenarios resulted in predictable outcomes, reducing the effectiveness of the training sessions.

*But at some point it just became very repetitive and instead of getting something out of it, I just saw a pattern and how to fix it.*

*[Undergrad 46]*

Others experienced boredom due to the repetition of similar stories and themes.

*There were only so many scenarios or stories. And even though they were still very much applicable to my life, I could anticipate what those stories were gonna be and I just got bored.*

*[Grad 35]*

**Length.** Of the interview participants, 82% (18/22) commented on the length of the training sessions. Each microdose session consisted of 10 CBM-I scenarios, along with pre-and post- intervention questionnaires. Researchers estimated that completing one microdose session would require roughly 5–10 minutes. Among the 18 participants that discussed session length, the majority (13/18, 72%) found the training sessions to be an appropriate length, although some expressed a preference for shorter sessions on occasion.

*I would've liked for it to be shorter, but at the same time I can understand that the longer you're thinking about it, the more realistically and detailed you can imagine your situation. And with this topic, you really kind of should be spending a little bit more time on it than you want to. So I think it was an appropriate length, even though at times it was not fun.*

*[Staff 41]*

*I thought they were a good length because I didn't have to worry about starting one and not being able to finish it.*

*[Undergrad 35]*

However, 28% (5/18) of participants that discussed length considered them too long, potentially impacting engagement.

*The session was just a bit long and I think maybe that was one reason that kept me from doing it as often. I just dreaded doing it because maybe it was too long.*

*[Grad 26]*

For comparison/triangulation, a survey question regarding session length was administered to all participants during their 2nd week of study participation. Participants were asked how they felt about the session length and 92 responses were received with 50% (46/92) of

participants indicating that the sessions were of the right length and 45% (41/92) of participants indicating that the sessions were too long (see Table 3).

**Write-your-own.** Of the interview participants, 64% (14/22) of interview participants provided feedback on the write-your-own scenario prompt. Some participants enjoyed the write-your-own scenario because it gave them a chance to apply their learning to real life situations.

*I think that was my favorite part because it kind of forced me to think a little more deeply about the modules I was doing rather than just answering the questions.*

*[Undergrad 35]*

Conversely, others found the write-your-own scenario to be frustrating because it took too long or felt redundant. Additionally, concerns about privacy surfaced, particularly among faculty and staff members, regarding inputting personal stories into the app.

*Crafting my own stories just irritated me. . .the homework aspect of it felt icky. But then also I didn't really want to because this is where I work. I didn't want to put scenarios in there. . .I think the exercise of thinking through my own scenarios would've been helpful if it wasn't writing them into the app.*

*[Staff 39]*

### Theme 3: Barriers to using the App

**Lack of understanding of intervention format or rationale.** Among the interview participants, 55% (12/22) expressed a lack of understanding of how to engage with the training sessions or the purpose of the intervention. Participants were given the flexibility to choose when and how many training sessions they completed on a given day, although it was recommended that they complete at least 2 microdose sessions. However, some participants found the structure of the app and study expectations to be ambiguous. Similarly, some participants were confused about the nature of the CBM-I training and did not understand the purpose of the tasks they were being asked to complete or how to engage with the stories.

*And at least at the beginning of the study I was like, what am I required to do? What are the tasks for today?*

*[Grad 52]*

*And so the very first survey, which was kind of longer, I thought, wait, is this like a reading comprehension survey? Like I really didn't know what it was. And so I would read the scenario and then it would say essentially, what do you think would be happening next? And I'm like, I don't know.*

*[Faculty 8]*

After each training scenario, participants were asked questions about the story and were supposed to answer based only on information given in the scenario. The purpose of this was to practice thinking in new ways about different situations. However, some participants found it unclear whether they were expected to answer questions based on information given in the story or based on their current experiences. This lack of understanding led some to feel confused when their answers were marked as incorrect.

*I just do not understand why it would say wrong when both of them [answer choices] could be logical, and even sometimes I chose one that seemed like the thing that I would make happen or that was happening in my life, and it'd be wrong. . .why am I being told I'm wrong?*

*[Faculty 1]*

**Forgetfulness.** Of the interview participants, 55% (12/22) cited forgetfulness as a reason for not using the app.

*I would usually just forget. . .I'm like already stressed out and then I'd scroll past it [the app], and I'm like, oh my god, I've forgotten to do that.*

*[Undergrad 25]*

Although the app did send push notifications, some participants encountered issues receiving notifications, which contributed to forgetfulness.

*If I don't get the notification on my phone and if I'm out, I'm not going to remember this.*

*[Grad 31]*

Some participants who mentioned not using the app due to forgetfulness expressed a preference for different types of notifications, such as calendar reminders rather than push notifications.

*It would've been better and more useful if it [reminders] would just populate [in my phone calendar] because I live or die by what's in my calendar. And if it's not on my calendar, I forget because there's just so much that's going on all day long.*

*[Staff 39]*

## Theme 4: Use patterns

**Used routinely.** Of the interview participants, 68% (15/22) used the app as part of their daily routine. These individuals found it most helpful to stick to a consistent schedule of using the app at the same time every day, rather than varying the time day by day or using the app specifically during moments of stress or anxiety.

*I mostly did it at the same time each day just because of how my schedule works.*

*[Faculty 8]*

*I would do it in the morning, and then right before I went to bed. I don't really remember doing it specifically if I was super anxious about something*

*[Grad 41]*

**Used when reminded.** Similarly, 50% (11/22) of interview participants used the app because they received a notification or reminder.

*I also liked that the app would send reminders, like you could set the time for every day that it would send reminders and as someone who also has been diagnosed with ADHD, like that's the only way I can get work done is if I have reminders.*

[Grad 35]

*I liked the feature where you could set a reminder to do them [training sessions] because that was like how I remembered to do it most days.*

[Undergrad 28]

**Chosen academics/work/career development domain.**   Participants also shared which stressor domain they used most often. Of the 8 domains to choose from, 82% (18/22) of interview participants stated they found the academics/work/career domain to be most important.

*I feel like I chose the work one most of the time because that is honestly where most of my anxiety comes from.*

[Staff 45]

*I tried to hit them all [domains], but the biggest source of stress in my life did happen to be graduate school. . .And so the graduate school one is definitely where I focused.*

[Grad 11]

## Theme 5: Suggestions for improvement

**Feature suggestions.**   Of the interview participants, 86% (19/22) suggested additional features to improve the apps functionality or enhance engagement. Among those participants, some suggested a progress tracking feature that would allow them to view their pending and completed tasks.

*If you want people to take three surveys a day or something, you can show like a little Domino's pizza loading bar and be like, you've already done 33%, you have two more left. Or like a checkbox or something for people to tangibly be like, yeah, I did this.*

[Staff 41]

*One thing that I just kind of thought of is I used the Duo Lingo app to help me to practice my French, and it tells me how well I've done recently. And if I have a streak of days in a row, it lets me know, hey, you've got this many days in a row. And it wasn't always clear to be able to see what my activity [in the HTC app] had been recently.*

[Faculty 4]

Others focused more on enhancing the customization and personalization of the app through tailored scenario content, adjustable training lengths, and flexible notifications. For example, some participants noted that certain scenario content was not relatable to their specific life stage.

*I don't know if you can collect some sort of information on the individual who's taking it, but a lot of times, it [the scenarios] would be something like, you're going on a Tinder date, and I'm married with two kids. That's pretty hard for me to imagine.*

[Staff 41]

Consequently, these participants often suggested personalizing the content by filtering scenarios specific to their demographic to enhance relevance.

*If there was some way to like screen participants and tailor the questions, I don't know, but if it was tailored to something that was more relatable to each participant, I feel like they could get more out of this program.*

*[Undergrad 46]*

Additionally, some expressed a desire to customize the length of the training sessions.

*I hoped it was longer when I had a bad day. And I hoped it was shorter when I was having a busy day. Maybe it can be helpful if I can adjust the length.*

*[Grad 13]*

Lastly, some participants suggested enhancing the notification system and expressed wanting more customizable notification preferences, including frequency and delivery method.

*I wish that just the average number of notifications was a little lower. And maybe people can opt into more or opt out of more.*

*[Faculty 4]*

*Maybe something that should be integrated is emails along with the notifications. . .if you're on your computer and not on your phone, maybe the different notifications would help.*

*[Grad 41]*

**Domain suggestions.** Finally, 73% (16/22) of interview participants suggested additional domains or ways to improve the existing ones. Of those 16 participants, 4 (25%) provided specific suggestions to improve the discrimination domain. The discrimination domain differed from the other 7 domains in that it did not provide specific scenarios for CBM-I training. Instead, it provided tools and resources for: 1) reporting discrimination or harassment, 2)self-care and wellness, 3) allyship (supporting others who have experienced discrimination, marginalization, or harassment), 4) community support, and 5) activism. Some participants appreciated how this domain was set up, while others mentioned wishing that this domain offered CBM-I training, similar to the other domains.

*I think I clicked on the discrimination topic one time. . .but it just brought me straight to links for like resources. So I think expanding that topic could be good because with more and more DEI conversations and stuff like that, people get really uncomfortable talking about it. So I think having something more in that area, like the resources were good and I'm grateful that they were there, but I think having something similar where you could kind of work out your ideas there too.*

*[Staff 41]*

*But if I click that category, then it shows that there's like this, this, and this kind of resource, rather than showing up the scenarios like it did for the other categories, and I felt that a little unhelpful. . .for instance, I always struggle to think that my English is not a problem, and that's also true, but like sometimes when I speak some wrong sentences, I become very small*

*and start to be very conscious about that. But people around me are more likely to be friendly and do not really care about that. So I hope there was some scenarios that helped me think that like all of their reactions are not really a discrimination. I don't, I know it can be very tricky, like it can be the app justifying there's no discrimination, but still I want something that can help or support mentally.*

*[Grad 13]*

Others suggested additional resources and tools within the discrimination domain that they wished were provided. This included adding resources specific to political and religious discrimination.

*The discrimination that I faced that made me go on was political discrimination. . .So I went on there to see if I could get any help with that. But the resources I don't think were really applicable to me in that sense.*

*[Grad 26]*

*I don't know if you guys had something about religion. . .I think that's an important aspect. Maybe that could go with the discrimination.*

*[Grad 49]*

Additional content recommendations were proposed for other domains. These include incorporating more scenarios tailored to an older demographic, such as addressing anxiety related to retirement, making friends later in life, dealing with a child's college rejection, coping with the loss of a family member, or managing the estate of a deceased parent.

In addition to suggestions made to the existing domains, other participants provided suggestions to add new domains, such as a domain specific to international students.

*I don't recollect right now but I don't know if there was a separate section that was directed towards international students. . .because there's so many international students and we have like our own set of problems, which sort of loosely relate to discrimination, but then also being homesick, being away from family, which weren't there.*

*[Grad 35]*

## Discussion

This study used mixed methods to explore users' experiences with the HTC platform and collect feedback on acceptability and ways to improve the app, which may also inform future designs of DMHIs more broadly. Qualitative data was collected via user feedback interviews and quantitative data via user experience questionnaires. Specifically, user experience questions regarding the effectiveness of the training program, the relatability of the training scenarios, and the length of the training sessions were analyzed to help validate and enrich the qualitative results (see Table 3). Data was also analyzed by demographic/university role breakdown. A triangulation-based framework was adopted to compare the quantitative results with the corresponding qualitative data. When examining the perceived length of the program based on both quantitative and qualitative data, no demographic group disproportionately favored one response over the other groups. Instead, the general consensus was split between

those who found the program to be the right length and those who considered it too long, as indicated in both the qualitative and quantitative findings. In regard to the program effectiveness, quantitative findings indicate that faculty tend to rate the program less positively than the other groups, finding it the least helpful, relatable, and believable. In contrast, staff and graduate students generally rate the program more positively, being more likely to find it helpful, relatable, and believable. Undergraduates fall somewhere in between (see S1 Table). Qualitative data align with these quantitative findings, as all faculty interviewees described the program as unrelatable, and none reported an increase in flexible thinking skills. Meanwhile, the majority of graduate interviewees (8/10, 80%) found the scenarios relatable and noted an increase in flexible thinking skills. These findings suggest a relationship between the relatability of the scenarios and the perceived effectiveness of the program. Overall, both quantitative and qualitative results indicate that the current version of HTC may be best suited for graduate students, and further work is needed to improve the program for faculty.

Among those who did not perceive the HTC program as effective, there seemed to be a gap in the relatability of content and alignment with needs/goals, potentially influencing their perceived efficacy. For example, some participants expressed not experiencing significant stress or anxiety prior to using the app, which could explain their limited observations of changes in their stress levels. Given participants were not required to meet a specific anxiety threshold to join the study, it is possible that other participants had similar experiences, though the mean baseline OASIS score indicated significant anxiety on average (a score of 8 is typically considered a clinical cutoff where most users would meet criteria for an anxiety disorder diagnosis and our sample mean was ~6.5). Moreover, one participant reported having other mental health conditions outside of anxiety that the app is not targeted to treat. Some participants also reported a lack of comprehension regarding the intervention's purpose, and a few stated they downloaded the app expecting it to be more meditative in nature. These findings underscore the importance of providing a clear introduction to the program's format and rationale during enrollment, allowing the user to understand the overall structure of the app and determine if it is a good fit for them. However, it is important to note that HTC does provide written and video explanations regarding the program rationale, learning goals, and ways to use the app during the first training session. This raises questions about whether users are effectively absorbing the introductory content or if changes are needed to present the program overview more effectively. Further work is encouraged in this area.

While many users did report perceived benefits from the program, qualitative interviews were conducted with an emphasis on gathering feedback for future app versions. Therefore, this discussion primarily centers on suggestions to enhance future iterations of HTC and DMHIs more broadly. Three main barriers to engaging with the platform were identified: contextual, personal, and technical. Contextual barriers arose from participants' lack of understanding of the intervention's rationale and how to engage with the program, hindering their ability to use it effectively. To address this, providing clear instructions on program expectations and guidelines for recommended use in multiple locations may enhance user engagement, as this information often appears to be overlooked or skimmed. Consistent with prior qualitative research, insufficient task explanations have been shown to deter users from continuing with digital health interventions [65]. Therefore, it is essential to present instructions clearly, and it may help to use various mediums [66]. HTC incorporates text-based and video instructions; to enhance clarity regarding task expectations, our users specifically suggest implementing a daily progress tracker, which could include gamification elements that might increase user motivation and engagement [67] while also clarifying what tasks need to be completed each day.

Personal barriers to engaging with the app included individuals citing forgetfulness as a reason for not participating in the program. This issue is directly linked to the technical barrier of notification problems, which is particularly important given half of the interviewees reported struggling with forgetfulness. Many participants indicated that they used the program specifically when reminded and appreciated the ability to schedule notifications at specific times. However, those who encountered issues with notifications stated that they only appeared when the app was open or running in the background. If the app was completely closed, reminders did not come through. This problem seemed specific to certain app versions and/or phone operating systems. Given that forgetfulness is a significant barrier to using the app, these technical issues likely impacted overall user engagement. To enhance notification customization and address both personal and technical barriers, we recommend offering users a range of notification delivery options, including push notifications, text messages, emails, or calendar reminders. Different notification strategies in DMHIs come with unique benefits and drawbacks that depend on user preferences. For instance, users with privacy concerns may prefer app notifications over text or email reminders, while those experiencing notification fatigue might opt for alternative systems, such as calendar integration. Given the pros and cons vary for each individual, it's preferable to offer a range of options. Additionally, allowing users to choose the frequency of notifications may also be beneficial. One study found that college students receive an average of 260 notifications per day, reflecting their high level of digital engagement [66]. Despite this extensive digital exposure, research shows that younger internet users are less affected by stress, burnout, depression, and anxiety related to communication load compared to those over age 50 [68]. Therefore, not only should notification types be customizable, but the frequency of notifications may also vary in effectiveness depending on age. Other research has shown that sending notifications containing tailored messages may also increase notification engagement [69]. In summary, expanding notification options and allowing users to choose what to opt into or out of will likely boost engagement and foster a sense of autonomy. However, with increased customization options, it's essential to include recommended default settings at baseline [67].

Consistent with other studies, our findings also suggest increasing app personalization would be particularly beneficial in the context of CBM-I scenario content, as some users expressed dissatisfaction with the relatability of the stories [32,70]. Given CBM-I relies on users imagining themselves in various situations to promote more flexible thinking, creating more personally relevant content may enhance users' ability to engage with the exercises more effectively. Therefore, we suggest future research should explore increasing personalization by using demographic information to filter content based on factors such as age, relationship status, and occupation, or assessing the types of interactions or places where each person tends to experience anxiety, or the particular unhelpful interpretations they tend to make. We would also like to highlight that despite our efforts to recruit faculty members, engaging them proved challenging, resulting in a small sample size for this study. However, it is worth noting that all three faculty participants indicated that the study content was not relatable. They also rated the app as the least helpful in reducing anxiety on the user experience questionnaire compared to the other groups, which may be linked to their perception of the content (see S1 Table). Therefore, we believe that increasing app personalization could be particularly beneficial for this demographic. Additionally, given that faculty members felt the content was tailored towards a younger audience, we strongly recommend involving more faculty in the co-design process for future iterations to ensure the content is relatable.

Similarly, some participants expressed the desire to customize content type and length. For example, the HTC program provided three CBM-I training scenario types: long scenarios, short scenarios, and write-your-own scenarios. Some participants reported receiving a write-

your-own training scenario, but not having the time or bandwidth to complete it at the moment, leading them to close the app entirely. They wished for the option to skip or choose certain scenario types as needed. Additionally, some participants expressed a desire to customize the length of microdose sessions, stating that sometimes they wanted to complete more or fewer than 10 training scenarios in one session. Although allowing participants to customize the type and length of training sessions may increase app engagement and user satisfaction, there is a concern that increasing customizability of training scenario type and length may sacrifice program effectiveness (e.g., if users reliably select such brief training periods that the dose is insufficient to reduce their engrained anxious thinking). Additionally, other research has found that too many customizable features may be burdensome and frustrating for people with motivational and cognitive challenges [71]. Instead, a focus on low-burden customization can enhance user autonomy and engagement with mental health interventions. For instance, studies indicate that daily checklists or progress tracking serve as effective, low-burden customizations that can enhance a sense of control and satisfaction [72,73].

To balance customization, user engagement, and app efficacy, it's essential to implement guided customization that provides users with a limited set of meaningful options, enhancing autonomy while preserving app integrity. In developing a DMHI, involving users in the design process is crucial to identify the most relevant customizable features for that demographic group. Additionally, as customization options are introduced, ongoing assessment of user satisfaction, engagement, and program efficacy is vital for understanding the trade-offs involved. Further work is needed to determine the best balance between increasing training flexibility and maintaining intervention effectiveness (e.g., ensuring users complete an adequate dose of training).

Last, offering more diverse training content is likely important for increasing engagement and accessibility. Research indicates that repetitive training content can lead to boredom and disinterest [70,74]. Incorporating a wider variety of scenario content and formats, including those suggested by users, such as scenarios relevant to international students, may improve acceptability and adherence. We recommend continuing to involve users in the co-design process for future iterations of HTC, particularly to develop more relevant scenarios for marginalized communities. This approach ensures that the content remains engaging and inclusive [75].

Our findings indicate that HTC is perceived as somewhat acceptable and effective at reducing stress and anxiety, with higher rates of perceived effectiveness among graduate students. Further research is needed to explore the general acceptability of DMHIs across demographic groups. However, the results suggest potential clinical implications, including the possibility of adding HTC to other forms of mental health care to support users in daily life (e.g., providing help managing stressful situations in between therapy sessions or medication check-ins). Alternatively, for users with minimal symptoms, using HTC may help manage symptoms so they do not escalate and reduce the need for (resource-intensive) care with a mental health professional. Additionally, HTC could be a valuable addition to support patients who are on waitlists so they have some support and symptoms do not worsen. Each of these possibilities requires further empirical evaluation, but given the difficulties accessing care and widespread need, DMHIs hold promise for increasing access to care.

At the same time, barriers such as forgetfulness and a lack of understanding regarding how to engage with the training sessions or the purpose of the intervention were noted. To address these challenges and optimize clinical impact, we recommend providing support and establishing regular check-ins when implementing DMHIs in a clinical setting. These check-ins could assess how users are interacting with the app, address any barriers they encounter, and offer encouragement or support as needed to help sustain user motivation and adherence to the program.

## Limitations

Several limitations should be considered in interpreting this interview and survey feedback. In some cases, individual qualitative interviews of participants took place a few months after their initial use of the intervention. This gap may have biased participants' reflections regarding their experience with HTC or its effectiveness. To address this, future iterations of the intervention could schedule post-treatment interviews more promptly after the completion of the program to gather participants' more immediate impressions of the program, as well as some interviews later to determine longer term impacts of the program.

Furthermore, while the sample size is consistent with related studies in the literature [20,30], it likely does not sufficiently represent the experience of all participants, especially considering the potential for sampling biases among the subset of users who agreed to take part in the post-intervention interviews. Specifically, we did not achieve the desired sample size for faculty, and this underrepresentation may limit our ability to draw robust conclusions. To address this, future research should prioritize targeted recruitment efforts for faculty, as they were identified as the most challenging demographic to engage. Additionally, most participants identified as White, non-Hispanic, and female, which means our results may not accurately represent the experiences of individuals from other racial, ethnic, and gender backgrounds.

Additional limitations include factors that may impact engagement, such as technical issues and study compensation. Participants reported technical issues, specifically regarding notifications, which hindered their ability to receive app reminders and likely affected their engagement. To address this limitation, future research should consider expanding notification delivery options to ensure participants receive alerts in a more accessible and effective format. Additionally, while participants received a $50 compensation for their involvement, we cannot determine how engagement might have differed without this incentive.

Moreover, we did not screen users for elevated anxiety before participation, so some participants did not meet a standard cutoff for clinical anxiety. This could skew participants' perception of the app's effectiveness if they do not struggle with anxiety or if they struggle with a different mental health condition that the app is not targeted to treat. In the future, screening for individuals who are high in trait, social, or generalized anxiety could provide a more appropriate sample to study the program's effectiveness (though we also value letting users choose whether they believe the program is something they want and need, as we did in the current study). We also did not require participants to stop other interventions while using HTC; this was done to increase external validity but we do not know how the experience of using HTC varies based on whether it is used on its own or as an adjunct to other forms of care. Finally, while the coders had varying levels of familiarity with HTC and considered diverse perspectives in the interviews, there is a possibility of reflexivity within the coding process.

## Conclusions

This study used mixed methods to explore users' experiences with the HTC platform, collecting feedback on its acceptability and suggestions for improvements, which could inform design of DMHIs more broadly. Qualitative data were gathered through user interviews, while quantitative data were collected via user experience questionnaires, specifically analyzing the program's effectiveness, relatability of scenarios, and opinions on session length. A triangulation-based framework was employed to compare the quantitative and qualitative findings, mostly demonstrating agreement between the methods. Faculty rated the program less positively than other groups, while graduate students found it the most effective and relatable. Undergraduate students and staff rated it moderately positive with somewhat more mixed

results. The results highlighted a gap in content relatability for some users, which may have influenced their perceptions of the app's efficacy. Recommendations for future iterations include clearer introductory content, enhanced personalization of training scenarios, and improved notification systems to address user engagement barriers. Overall, while HTC showed promise in reducing stress and anxiety, further research is needed to understand its acceptability across diverse demographic groups and to refine the program for optimal effectiveness.

## Supporting information

**S1 Fig. Thematic Maps (initial and revised to only include the most saturated codes).**
(PDF)

**S1 Text. Standards for Reporting Qualitative Research (SRQR) Checklist.**
(PDF)

**S2 Text. Semi-structured Interview Guide.**
(PDF)

**S1 Table. Results from Questionnaire by Group.**
(DOCX)

## Author Contributions

**Conceptualization:** Sarah Livermon, Audrey Michel, Yiyang Zhang, Emma Toner.

**Data curation:** Sarah Livermon, Kaitlyn Petz.

**Formal analysis:** Sarah Livermon, Audrey Michel, Yiyang Zhang.

**Funding acquisition:** Laura E. Barnes, Bethany A. Teachman.

**Investigation:** Sarah Livermon, Audrey Michel, Yiyang Zhang.

**Methodology:** Sarah Livermon.

**Project administration:** Sarah Livermon, Emma Toner.

**Software:** Mark Rucker, Mehdi Boukhechba.

**Supervision:** Bethany A. Teachman.

**Visualization:** Sarah Livermon, Audrey Michel, Yiyang Zhang.

**Writing – original draft:** Sarah Livermon, Audrey Michel, Yiyang Zhang.

**Writing – review & editing:** Kaitlyn Petz, Emma Toner, Laura E. Barnes, Bethany A. Teachman.

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
