## [Decision Letter · Decision Letter 0]

3 Sep 2024

PDIG-D-24-00309

A Mobile Intervention to Reduce Anxiety Among University Students, Faculty, and Staff: Mixed Methods Study on Users’ Experiences

PLOS Digital Health

Dear Dr. Livermon,

Thank you for submitting your manuscript to PLOS Digital Health. After careful consideration, we feel that it has merit but does not fully meet PLOS Digital Health's publication criteria as it currently stands. Therefore, we invite you to submit a revised version of the manuscript that addresses the points raised during the review process.

Please submit your revised manuscript within 60 days Nov 02 2024 11:59PM. If you will need more time than this to complete your revisions, please reply to this message or contact the journal office at digitalhealth@plos.org. Please include the following items when submitting your revised manuscript:

We look forward to receiving your revised manuscript.

Kind regards,

Haleh Ayatollahi

Section Editor

PLOS Digital Health

Additional Editor Comments (if provided):

Reviewers' comments:

Reviewer's Responses to Questions

**Comments to the Author**

1. Does this manuscript meet PLOS Digital Health’s publication criteria? Is the manuscript technically sound, and do the data support the conclusions? The manuscript must describe methodologically and ethically rigorous research with conclusions that are appropriately drawn based on the data presented.

Reviewer #1: Yes

Reviewer #2: Partly

Reviewer #3: Yes

2. Has the statistical analysis been performed appropriately and rigorously?

Reviewer #1: Yes

Reviewer #2: No

Reviewer #3: Yes

3. Have the authors made all data underlying the findings in their manuscript fully available (please refer to the Data Availability Statement at the start of the manuscript PDF file)?

Reviewer #1: Yes

Reviewer #2: No

Reviewer #3: Yes

4. Is the manuscript presented in an intelligible fashion and written in standard English?

Reviewer #1: Yes

Reviewer #2: Yes

Reviewer #3: Yes

5. Review Comments to the Author

Reviewer #1: Dear authors,

I would like to offer some constructive feedback to enhance the clarity, impact, and breadth of your manuscript.

1. Effectiveness and Engagement Interpretation

The manuscript does a commendable job of analyzing the effectiveness of the Hoos Think Calmly app. However, there is a slight tendency to overestimate the intervention’s positive impact on anxiety reduction. Given that the majority of participants rated the app as only "somewhat helpful," a more balanced interpretation of these findings would strengthen the manuscript’s conclusions.

Similarly, the manuscript emphasizes user engagement, but the reliance on reminders and the issues with the notification system suggest that actual engagement might be lower than reported. Revising the discussion to reflect these nuances would provide a more accurate representation of user interaction with the app.

2. Relatability and Personalization Concerns

The issue of content relatability, particularly among older users and faculty, is significant. While this is acknowledged in the manuscript, its potential impact on the effectiveness of the intervention might be underemphasized. Strengthening the discussion around this point and offering more concrete suggestions for addressing this issue (e.g., tailoring content to different demographic groups) would be beneficial.

3. Technical Barriers

The technical barriers, particularly the inconsistency of push notifications, are briefly mentioned. However, these barriers are likely to have a more substantial impact on user engagement than currently reflected. A more detailed exploration of these technical challenges and their potential solutions would enhance the manuscript’s practical relevance.

4. Methodological Considerations

While the methodological rigor is evident, the decision to code transcripts by a single coder after establishing high inter-coder reliability diverges from the original pre-registration. While this is understandable, a more transparent discussion of this change and its implications for the findings would be valuable.

5. Proportion of Self-Citations

It is noted that a considerable proportion of the references are self-citations related to your previous work on the MindTrails website and Cognitive Bias Modification for Interpretation (CBM-I). While it is reasonable to reference foundational studies, I encourage you to ensure that the manuscript is well-balanced by incorporating a wider range of studies from other researchers. This will not only broaden the context but also strengthen the manuscript’s engagement with the broader literature.

Thank you for your attention to these matters, and I appreciate your efforts in advancing this important area of research.

Best regards,

Reviewer #2: Dear Authors, 

The sample size of 22 is a poor one to claim anything. Although I like the research methodology, which is sound and acceptable to me. 

Please revise the paper with at least 500 samples to get something meaningful out of your work. 

I also suggest to read these two articles:

a) Chattopadhyay S, Ray P, Land L, Li J. 'A Framework for Assessing ICT Preparedness for e-Health 

Implementation', Accepted in IEEE 10th International Conference on e-Health Networking, Application 

& Services", (IEEEHealthcom2008) to be held in Singapore from 7 – 9

th July (2008) 

b)Chattopadhyay S., – “A Prototype Depression Screening Tool for Rural Healthcare: A Step towards 

e-Health Informatics”, Journal of Medical Imaging and Health Informatics (2012); 2(3): 244-249.

If you feel these are insightful and helpful to your work to describe, you may please cite it. 

Looking forward to receiving your revised version.

Reviewer #3: I've thoroughly reviewed your paper and offered constructive feedback to enhance its impact. Kindly consider my suggestions to improve clarity and effectiveness. Best wishes.

Abstract:

• Provide specific statistical data on the prevalence of anxiety in college communities. Include recent, peer-reviewed sources to support this claim. This will establish a more compelling rationale for the study's importance. 

• Briefly explain what CBM-I entails and how it differs from other digital mental health interventions.. 

• Specify the quantitative and qualitative methods used in the study. Explain how these methods complement each other and why this approach was chosen. 

• Provide more detail on how the thematic analysis was conducted. Mention the analytical framework used and how themes were identified and validated. 

• Include more specific data from the biweekly user experience questionnaires. Provide percentages or numerical data to support the statement about the program's helpfulness. This will add more substance to the quantitative findings. 

• Expand on how the findings contribute to the broader field of digital mental health interventions. Explain how the results can inform future development and implementation of similar applications. 

• Strengthen the concluding statement by clearly articulating the main takeaways from the study and their significance for both research and practice in digital mental health interventions. 

introduction:

1. Expand the review of existing literature on digital mental health interventions (DMHIs) and Cognitive Bias Modification for Interpretation (CBM-I). Include more recent studies and meta-analyses to provide a comprehensive overview of the current state of research in this field. Critically analyze the findings of these studies to identify gaps in knowledge that your research aims to address.

2. Develop a more robust theoretical framework that underpins the study. Clearly articulate the theoretical principles behind CBM-I and how they relate to anxiety reduction. Explain the mechanisms by which CBM-I is hypothesized to work and how these mechanisms are incorporated into the "Hoos Think Calmly" (HTC) application.

3. Provide a more detailed justification for conducting this specific study. Explain why evaluating user experience and acceptability is crucial for the development and improvement of DMHIs. Elaborate on how the findings from this study will contribute to advancing the field of digital mental health interventions.

4. Offer a more comprehensive description of the HTC application. Include details about its design, functionality, and how it delivers CBM-I training. Explain how the app's features address the specific needs and challenges of the university community.

5. Include more recent references to ensure the literature review reflects the most current research in the field. Ensure all claims are properly supported by credible, peer-reviewed sources. Consider including a mix of seminal works and cutting-edge research to provide a comprehensive overview of the topic.

Method:

1. Expand on the ethical considerations section. Provide more details about the informed consent process, including how participants were informed about the study's purposes, risks, and benefits. Elaborate on measures taken to protect participant confidentiality and data security, particularly given the sensitive nature of mental health data.

2. Offer a more detailed explanation of the CBM-I training scenarios. Include information about how the scenarios were developed and validated. Discuss any pilot testing or iterations of the scenarios that occurred before the main study.

3. Offer more details about the semi-structured interview process. Describe how the interview guide was developed and piloted. 

4. Provide a more detailed account of the thematic analysis process. Explain how themes were identified, refined, and validated. Include information about how disagreements between coders were resolved and how the final themes were decided upon.

5. Expand on the analysis of the survey data. Describe any statistical software used and provide more details about the specific descriptive statistics calculated. Explain how missing data, if any, was handled.

6. Provide more information about the psychometric properties of the measures used (OASIS, PHQ-2, PSS-4). Include details about their validity and reliability in similar populations.

7. Provide a more robust justification for using a mixed-methods approach. Explain how the quantitative and qualitative data complement each other and how they will be integrated in the analysis and interpretation of results.

Results:

1. Conduct a more thorough analysis of how responses varied across different demographic groups (e.g., students vs. faculty, different age groups). Present these findings in a clear, systematic manner, potentially using subheadings for each demographic category.

Discussion:

1. Provide a more comprehensive synthesis of the mixed-methods results. Develop a framework or model that illustrates how the quantitative and qualitative findings complement each other and contribute to a holistic understanding of user experiences with HTC.

2. Elaborate on the potential benefits and challenges of increasing app personalization and customization. Discuss the trade-offs between user preferences and maintaining intervention fidelity. Provide specific recommendations for balancing these competing demands in future iterations of HTC and similar DMHIs.

3. Offer a more in-depth examination of the barriers to app engagement identified in the study. Categorize these barriers (e.g., technical, personal, contextual) and discuss their implications for app design, implementation, and user support strategies.

4. Provide a more comprehensive analysis of notification preferences and their impact on user engagement. Discuss the potential benefits and drawbacks of different notification strategies, and offer specific recommendations for optimizing notification systems in DMHIs.

5. Offer a more critical examination of the study's limitations. Discuss how these limitations might have influenced the findings and provide specific recommendations for addressing these limitations in future research.

6. Provide a more comprehensive analysis of the potential clinical implications of these findings. Discuss how the results might inform clinical practice, particularly in terms of recommending and supporting the use of DMHIs like HTC.

7. Provide a more impactful conclusion that synthesizes the key findings, their implications, and future directions. Offer a clear and compelling vision for the future development and implementation of HTC and similar DMHIs.

6. PLOS authors have the option to publish the peer review history of their article (what does this mean?). If published, this will include your full peer review and any attached files.

**Do you want your identity to be public for this peer review?** For information about this choice, including consent withdrawal, please see our Privacy Policy.

Reviewer #1: Yes: Rabie Adel El Arab

Reviewer #2: Yes: Subhagata Chattopadhyay

Reviewer #3: Yes: Roghieh Nooripour

---

## [Decision Letter · Decision Letter 1]

18 Nov 2024

A Mobile Intervention to Reduce Anxiety Among University Students, Faculty, and Staff: Mixed Methods Study on Users’ Experiences

PDIG-D-24-00309R1

Dear Livermon,

We are pleased to inform you that your manuscript 'A Mobile Intervention to Reduce Anxiety Among University Students, Faculty, and Staff: Mixed Methods Study on Users’ Experiences' has been provisionally accepted for publication in PLOS Digital Health.

Best regards,

Haleh Ayatollahi

Section Editor

PLOS Digital Health

**Additional Editor Comments (if provided):**

**Reviewer Comments (if any, and for reference):**

Reviewer's Responses to Questions

**Comments to the Author**

1. If the authors have adequately addressed your comments raised in a previous round of review and you feel that this manuscript is now acceptable for publication, you may indicate that here to bypass the “Comments to the Author” section, enter your conflict of interest statement in the “Confidential to Editor” section, and submit your "Accept" recommendation.

Reviewer #1: All comments have been addressed

Reviewer #3: All comments have been addressed

2. Does this manuscript meet PLOS Digital Health’s publication criteria? Is the manuscript technically sound, and do the data support the conclusions? The manuscript must describe methodologically and ethically rigorous research with conclusions that are appropriately drawn based on the data presented.

Reviewer #1: Yes

Reviewer #3: Partly

3. Has the statistical analysis been performed appropriately and rigorously?

Reviewer #1: Yes

Reviewer #3: Yes

4. Have the authors made all data underlying the findings in their manuscript fully available (please refer to the Data Availability Statement at the start of the manuscript PDF file)?

Reviewer #1: Yes

Reviewer #3: Yes

5. Is the manuscript presented in an intelligible fashion and written in standard English?

Reviewer #1: Yes

Reviewer #3: Yes

6. Review Comments to the Author

Reviewer #1: Thank you for addressing the comments

Reviewer #3: I have reviewed the revised manuscript and can confirm that the authors have addressed the revisions appropriately. The manuscript now appears suitable for publication in its current form.

7. PLOS authors have the option to publish the peer review history of their article (what does this mean?). If published, this will include your full peer review and any attached files.

**Do you want your identity to be public for this peer review?** For information about this choice, including consent withdrawal, please see our Privacy Policy.

Reviewer #1: **Yes: **Rabie Adel El Arab

Reviewer #3: **Yes: **Roghieh Nooripour
